# Dissecting new genetic components of salinity tolerance in two-row spring barley at the vegetative and reproductive stages

Stephanie Saade[1]*, Chris Brien[2,3,4], Yveline Pailles[1], Bettina Berger[2,4], Mohammad Shahid[5], Joanne Russell[6], Robbie Waugh[2,6,7], Sónia Negrão[8], Mark Tester[1]

1 Biological and Environmental Sciences and Engineering (BESE), King Abdullah University of Science and Technology (KAUST), Thuwal, Saudi Arabia, 2 School of Agriculture, Food and Wine, Waite Research Precinct, University of Adelaide, Urrbrae, South Australia, Australia, 3 School of Information Technology and Mathematical Sciences, University of South Australia, Adelaide, South Australia, Australia, 4 The Plant Accelerator, Australian Plant Phenomics Facility, Waite Research Precinct, University of Adelaide, Urrbrae, South Australia, Australia, 5 International Center for Biosaline Agriculture (ICBA), Dubai, United Arab Emirates, 6 Cell and Molecular Sciences, The James Hutton Institute, Invergowrie, Dundee, Scotland, 7 Division of Plant Sciences, School of Life Sciences, University of Dundee at The James Hutton Institute, Invergowrie, Dundee, Scotland, 8 School of Biology and Environmental Sciences, University College Dublin, Belfield, Dublin, Ireland

* stephanie.saade@kaust.edu.sa

**Data Availability Statement:** All relevant data are within the paper and its Supporting Information files.

## Abstract

Soil salinity imposes an agricultural and economic burden that may be alleviated by identifying the components of salinity tolerance in barley, a major crop and the most salt tolerant cereal. To improve our understanding of these components, we evaluated a diversity panel of 377 two-row spring barley cultivars during both the vegetative, in a controlled environment, and the reproductive stages, in the field. In the controlled environment, a high-throughput phenotyping platform was used to assess the growth-related traits under both control and saline conditions. In the field, the agronomic traits were measured from plots irrigated with either fresh or saline water. Association mapping for the different components of salinity tolerance enabled us to detect previously known associations, such as *HvHKT1;5*. Using an "interaction model", which took into account the interaction between treatment (control and salt) and genetic markers, we identified several loci associated with yield components related to salinity tolerance. We also observed that the two developmental stages did not share genetic regions associated with the components of salinity tolerance, suggesting that different mechanisms play distinct roles throughout the barley life cycle. Our association analysis revealed that genetically defined regions containing known flowering genes (*Vrn-H3*, *Vrn-H1*, and *HvNAM-1*) were responsive to salt stress. We identified a salt-responsive locus (7H, 128.35 cM) that was associated with grain number per ear, and suggest a gene encoding a vacuolar $H^+$-translocating pyrophosphatase, *HVP1*, as a candidate. We also found a new QTL on chromosome 3H (139.22 cM), which was significant for ear number per plant, and a locus on chromosome 2H (141.87 cM), previously identified using a nested association mapping population, which associated with a yield component and interacted with salinity stress. Our study is the first to evaluate a barley diversity panel for salinity stress under both controlled and field conditions, allowing us to identify contributions from

**Funding:** The research reported in this publication was supported by funding from KAUST to the Baseline of Mark Tester. https://saltlab.kaust.edu.sa.

**Competing interests:** The authors have declared that no competing interests exist.

new components of salinity tolerance which could be used for marker-assisted selection when breeding for marginal and saline regions.

## Introduction

Soil salinization is a limiting factor in crop production that affects at least 20% of irrigated lands, a number which is bound to increase due to poor irrigation practices and intrusions of groundwater caused by rising sea levels [1, 2]. While many crops grow or yield poorly in saline soils, barley (*Hordeum vulgare* L.) has been deemed the most salt-tolerant cereal crop [3]. Due to its resilience and relatively stable yields, barley is cultivated in both highly productive areas and subsistence low-input agricultural systems [4]. In 2016–2017, an estimated 150 million tons were produced globally for animal feed, human food, and as a raw material for the malt industry [5]. Since barley adapts well to stressed environments, it has been the target of a vast number of studies to predict the responses of crops to climate change, especially in terms of feeding an ever-increasing population and facing freshwater shortages [6], and to understand the mechanisms underlying its response to soil salinity.

Munns *et al.* [3] proposed three main mechanisms of salinity tolerance: (i) osmotic tolerance, i.e., "shoot ion-independent tolerance", (ii) ion exclusion from the shoot, and (iii) tissue tolerance [7]. To determine the genetic basis for barley's salinity tolerance, several forward genetics studies have explored the impact of salinity on these three mechanisms. The first of these, shoot ion-independent tolerance takes place in the early stage of the salt stress before sodium ions had the time to accumulate. During this rapid response of the plant to stress, growth is reduced [3]. The shoot ion-independent mechanism of tolerance is the least studied, because of the difficulties in phenotyping, but signaling type genes are thought to play a major role [8, 9]. A locus that contributes to shoot $Na^+$ concentration, *HvNax4*, was mapped in a doubled-haploid population on chromosome 1HL [10]. Fine mapping of the *HvNax4* locus proposed *HvCBL4*, the homolog of the Arabidopsis *salt-overly sensitive 3* (*SOS3*), as a candidate gene [11].

In terms of the ionic component of salinity tolerance, Hazzouri *et al.* [12] performed an association mapping study, where a USDA mini-core barley collection was screened for salinity tolerance under field conditions. A locus associated with leaf sodium content was identified on chromosome 4HL; the suggested candidate gene was *HKT1;5*, a high-affinity potassium transporter [12–14]. The increased salinity tolerance may be achieved by *HKT1;5* retrieving $Na^+$ from the xylem sap at the root level, which would prevent $Na^+$ from accumulating in the shoot [12, 15–17]. Other association studies using diversity panels have also been used to assess salinity tolerance in terms of biomass production, shoot and root ion contents, survival scores, and leaf chlorosis [18, 19].

Tissue tolerance, an important component of salinity tolerance [3, 20], is related to barley's ability to maintain photosynthetic activity and other vital functions in the presence of high levels of $Na^+$ in the shoot. This could be achieved by sequestering excess $Na^+$ into the vacuole, and a vacuolar $H^+$-inorganic pyrophosphatase (V-PPase) has been identified that could provide the required energy [21]. Indeed, a locus on chromosome 7HS called *HvNax3* explained up to 25% of the leaf sodium content in a bi-parental mapping population [22]; the suggested candidate gene underlying this locus was *HVP10*, a vacuolar $H^+$-translocating pyrophosphatase [23].

In addition to the three components advocated by Munns *et al.* [3], other physiological components have been hypothesized to play a role in salinity tolerance. For instance, Al-

Tamimi *et al.* [24] identified a locus in rice responsible for maintaining transpiration efficiency under salinity stress. A comprehensive framework of the genetics of salinity tolerance is described in Morton *et al.* [25].

In this work, we present an association mapping approach to identify loci that control components of salinity tolerance, with particular emphasis on identifying QTL for the difficult-to-study shoot ion-independent mechanism. A diversity panel comprised of 377 two-row spring barley accessions originating from Europe was phenotyped at The Plant Accelerator® (TPA) (Adelaide, Australia), and at the International Center for Biosaline Agriculture (ICBA; Dubai, UAE). To target shoot ion-independent tolerance, we used non-destructive high-throughput imaging during the vegetative stage and investigated the effects of salinity on growth rate. To study yield-related traits, we conducted two consecutive years of field trials at ICBA, irrigating plots with either non-saline or saline water. The location of the ICBA field trial was ideal for studying salinity tolerance during the reproductive stage because of its sandy soil and very low precipitation. The association panel was genotyped with the 9K iSelect SNP set [26] and QTL associated with salinity tolerance were identified using "classical" and "interaction" [24] mixed linear models (MLM).

## Materials and methods

We would like to state that the field trials described in this section were conducted at the International Center for Biosaline Agriculture (ICBA in Dubai, United Arab Emirates) under their permission.

### Plant material

A diversity panel of 377 two-row spring barley accessions originating from Europe was evaluated for salinity tolerance. A list summarizing all the accessions used in this study, country of origin, and year of release is available in S1 Table. Parts of the diversity panel have been described in other studies [26–30].

### Genotyping of the diversity panel

Genotyping of the 377 accessions was performed using the Illumina Infinium iSelect HD 9k chip, as described in Comadran *et al.* [26]. From the original set of 7,864 high-confidence, gene-based single nucleotide polymorphisms (SNPs), a total of 5,062 informative SNPs (polymorphic, minor allele frequency $\geq$ 5%, no heterozygosity, < 20% missing) were kept for population structure analysis in STRUCTURE. Duplicated SNPs were deleted to reduce the computational time, resulting in 4,226 unique SNPs for the association analyses. Imputation of the genotypic data was performed using the link.im function in the linkim package in R [31].

### Estimating the population structure

To investigate possible population stratifications, we analyzed the SNP data (5,062 SNPs) from the 377 accessions using the STRUCTURE software, which uses a Bayesian clustering approach to assign individuals to $K$ subpopulations [32]. Ten independent runs were performed, with $K$ = 1 to 6, 50,000 burn-in periods, and 10,000 Markov Chain Monte Carlo iterations for each value of $K$. We assumed that each individual belonged to only one population and used the default admixture model for the ancestry of individuals and the correlated allele frequencies. In order to choose the best $K$, the results were analyzed using the $\Delta K$ method [33], implemented in the STRUCTURE HARVESTER software [34]. The clusters were permuted and aligned across runs using CLUMPP [35]. Finally, the DISTRUCT software was used to

visualize the structure of the population by plotting the Q matrix [36], in which the accessions were sorted by their year of release (S1 Fig). The population structure was accounted for in the association models.

## Phenotyping at The Plant Accelerator®

**Experimental setup and salt application.**    The barley diversity panel was evaluated during the vegetative stage using continuous, non-destructive high-throughput phenotyping at TPA (Adelaide, Australia; −34.97113, 138.63989). The phenotyping experiment was conducted under controlled conditions between March 21, 2014, and April 23, 2014, in the northeast (NE) and northwest (NW) Smarthouses. Temperatures in these Smarthouses ranged between 24˚C during the day and 18˚C at night, with an average relative humidity of 73%. Four seeds per accession were sown 2 cm deep in a 2.5 L free-draining white pot. The soil mix consisted of 50% (v/v) University of California (UC) mix, 35% (v/v) cocopeat mix, and 15% (v/v) clay/ loam from Angle Vale (South Australia). At the two-leaf stage, the seedlings were thinned down to one plant per pot and evaluated for uniform size and developmental stage. White gravel (particle size 2–5 mm) was added to reduce water evaporation from the soil surface. A blue frame was set in the pot to provide support to the plants. At the emergence of the third leaf (13 days after sowing), the pots were placed in deep saucers and loaded onto conveyor systems. Imaging of the plants and watering up to 17% (w/w) gravimetric water content were performed daily by the Scanalyzer 3D system (LemnaTec GmbH, Aachen, Germany). At the emergence of the fourth leaf (17, 20, or 21 days after sowing due to variations in plant growth), the third leaf was marked and a salt treatment was applied (200 mL of 350 mM NaCl) to reach a concentration of 200 mM NaCl in the soil solution once the water content dried back down to 17% (w/w). Control plants were given an equal volume (200 mL) of water at the time of treatment. Then, the plants were imaged for 11 days after the salt treatment application. To evaluate the sodium and potassium contents in the shoot, the fourth fully expanded leaf of each plant growing under saline conditions was collected. This ensured that the collected leaf had been exposed to the same period of stress despite variations in plant growth across accessions. The leaves were dried overnight in an oven at 70˚C and the dry mass was recorded. The dried samples were digested in 10 mL of 1% (v/v) nitric acid for 4 hours at 80˚C and diluted with milliQ water to a dilution factor of 1:20. The sodium and potassium contents per dry mass were analyzed using a flame photometer (Model 420 flame photometer from Sherwood, UK).

**Experimental design.**    The phenotyping experiment was performed in the NE and NW Smarthouses, which each consisted of 24 lanes with 22 positions. In each position, there was a cart (a pot) containing a single plant. Because sets of four lanes were found to be homogeneous in terms of plant growth variability, each set of four lanes was grouped into one zone, resulting in six zones per Smarthouse [37]. We used a split-plot design (S2 Fig) to allocate the accessions and conditions, where a main plot corresponded to a pair of consecutive positions, and each position was a subplot that contained one control (no salt) and one salt-treated plant with the same barley accession. The use of the split-plot design minimized the effect of spatial variations between treatments because both the control and the treated plant from the same accession were located next to each other. Accessions were partially replicated (30%) to estimate spatial variation within a Smarthouse from the differences between main plots with replicated lines. The accessions were allocated to main plots using a "nearly-trend-free" block design, where each block corresponded to one of the six zones (S2 Fig). Salt and control conditions were randomized between the two subplots within each main plot. We used *DiGGer* [38], a package for the R statistical computing environment [39], to create the main-plot design, as it initially

randomizes the allocation of accessions among the main plots. *DiGGer* then iteratively rear-ranges allocation, taking into account sources of variation, so that the resulting distribution of the replicated accessions is not totally random.

**Daily imaging of the plants.** Daily imaging of the plants using the LemnaTec Scanalyzer 3D (LemnaTec GmbH, Aachen, Germany) started at 13 days after sowing at TPA. Two red-green-blue (RGB) images were captured from the side at a 90° rotation from each other (side view 1 and side view 2) and one RGB image was captured from above (top view). LemnaGrid software (LemnaTec GmbH, Aachen, Germany) was used to perform foreground-background separation and to remove noise from the images. Pixels identified as belonging to the plant in each of the three images were counted, and the sum of the three (side view 1 + side view 2 + top view) was used to calculate the projected shoot area (PSA). PSA was previously shown to linearly correlate with plant biomass at this growth stage in barley [40].

The PSA was processed using the smoothing and extraction of traits (SET) method described by Brien *et al*. [41] using *imageData* [42], a package for the R statistical computing environment [39]. The absolute growth rate (AGR) and relative growth rate (RGR) were calculated by taking the difference between the consecutive PSA and ln(PSA) values, respectively, and dividing this number by the time difference. After calculating PSA, RGR, and AGR, those plants which did not grow, were very slow-growing, had abnormal growth curves, whose control outperformed the salt-treated counterpart, or which flowered at that time of the experiment were excluded from subsequent analyses. After removing these plants, the PSA, RGR, and AGR values were smoothed by fitting cubic smoothing splines for each plant. After examining the plots of time-course data at TPA, we decided to investigate growth for three time-intervals: 1–4, 4–8, and 8–11 days after salt treatment (Fig 1).

The complete RGB dataset is publicly available through Zegami at TPA (https://zegami.plantphenomics.org.au/)

**Spatial correction of phenotypic data.** To generate phenotypic means adjusted for the spatial variation in the Smarthouses, a mixed-model analysis was performed for each trait using *ASReml-R* [43] and *asremlPlus* [44], packages for the R statistical computing environment [39] as described in Al-Tamimi *et al*. [24].

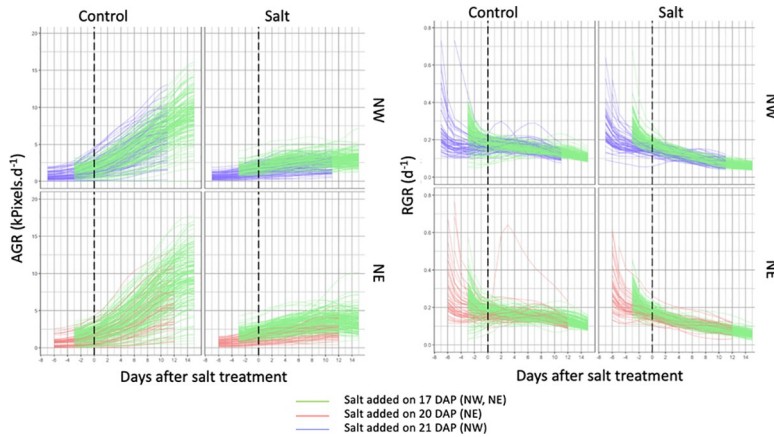

**Fig 1. Plots of smoothed absolute growth rate and relative growth rates.** Plots of absolute growth rate (AGR, right panels) and relative growth rate (RGR, left panels) smoothed by cubic splines for Northwest (NW, top panels) and Northeast (NE, bottom panels) Smarthouses. Different colors refer to different starting days of the salt treatment. Based on these plots, we decided to split the period of the experiment into the intervals: 1–4, 4–8, and 8–11 days after salt treatment. DAP refers to days after planting.

In brief, the formula of the maximal mixed model for this analysis is:

$$\mathbf{y} = \mathbf{X}\boldsymbol{\beta} + \mathbf{Z}\mathbf{u} + \mathbf{e},$$

where **y** is the vector of values of the trait analyzed and **β, u** and **e** are the vectors for the fixed, random, and residual effects, respectively. The design matrices corresponding to **β** and **u** are denoted by **X** and **Z**, respectively. The smarthouse, zone, position, treatment, and genotype effects were all accounted for in the model as described in detail in Al-Tamimi *et al.* [24]. From these analyses, the best linear unbiased estimates were obtained and used as an input for the subsequent association analysis. The heritability was calculated according to Cullis *et al.* [45].

## Phenotyping at the field site

A total of 377 European barley accessions, along with three salt-tolerant check lines (116/2A, 58/1A, CM72), were grown at ICBA in Dubai, United Arab Emirates (N 25˚ 05.847; E 055˚ 23.464), over the course of two years (2013/2014 and 2014/2015). The plots were irrigated with either non-saline (1dS/m, referred to as the control condition) or saline (17dS/m, referred to as the saline condition) water. A full description of the soil characteristics, fertilization and irrigation practices, and field design was detailed in Saade *et al.* [46]. The following agronomic traits were recorded under both the control and saline conditions: flowering time (HEA), maturity time (MAT), ripening period (RIP), plant height (HEI), ear number per plant (EAR), grain number per ear (GPE), dry mass per $m^2$ (DRY_WT), grain mass per $m^2$ (or yield, YLD), and harvest index (HI).

The phenotypic data was corrected for spatial variation using a multi-environment trial (MET) analysis in which four year-by-condition combinations were considered separate trials [47]. We examined variograms, following Gilmour *et al.* [48], to identify the environmental terms that were sources of variation and which needed to be added to the MET analysis model. The models were fitted in ASReml v3.0–1 [43] for R v3.2.0 [49]. The predicted means for the traits of each accession were corrected for spatial variation (S2 Table) and used as input for further phenotypic and association mapping analyses. The significance of the differences between the means of the two conditions was tested using analysis of variance (ANOVA), and the correlation among traits was performed using the Pearson correlation. Full details about the spatial correction of the field data can be found in Saade *et al.* [46].

## Association model analysis

To identify the loci associated with salinity tolerance, we performed an association mapping analysis on all 377 barley accessions, using predicted means adjusted for year and condition. We used the "classical method", an MLM in which we accounted for population structure and kinship among accessions. The principal component analysis (PCA.total = 3) and the kinship (kinship.algorithm = "EMMA") matrix were generated using the Genome Association and Prediction Integrated Tool package (GAPIT) [50]. The classical model (using GAPIT) was only used to examine the sodium and potassium leaf contents because the flame photometer data was only collected from plants under saline conditions.

We also used the "interaction method" that was described by Al-Tamimi *et al.* [24]. This MLM allowed us to examine the interaction between the SNP markers and the treatment, enabling the detection of genetic regions associated solely with the treatment. We included three principal components for population structure and relatedness between the individual (kinship) matrices, which were generated using GAPIT [50], as fixed effects in the model. The kinship was included as a random effect. We fitted the model in *ASReml* [43], a package for

the R computing environment [39]. For the field data, we also added the year of the experiment in the interaction term of the model as the experiments were carried out over two years.

To determine the significance of associations between the SNP markers and a given trait, we used the Bonferroni-adjusted threshold of α = 0.05. Results from the interaction-model association analyses are included in S3 Table. To facilitate discussion of the results, we report the position of the most significant SNP marker for each locus. Manhattan plots were generated using the R package *qqman* [51] (S3 Fig). To estimate the linkage disequilibrium between two SNPs on the same chromosome, we used the software PLINK [52] (http://zzz.bwh.harvard.edu/plink) to calculate the squared correlation ($r^2$) based on genotypic allele counts. S4 Table contains the SNPs with $r^2 > 0.8$ per chromosome.

## Results and discussion

In this work, we evaluated a collection of 377 two-row spring barley genotypes for salinity tolerance using high-throughput phenotyping at the vegetative stage and field trials at the reproductive stage. Previous works have combined high-throughput controlled and field phenotyping to identify genetic loci associated with stresses, other than salinity, in other plant species [53, 54]. However, to the best of our knowledge, this is the first association study that combines the two experimental settings to elucidate the genetic components of barley salinity tolerance.

The salt levels applied in the two experimental setups (200mM at TPA and 17dS/m ICBA; which are roughly equivalent) were considered appropriate for our studies because they significantly affected all studied traits without causing major premature plant senescence. At TPA, we observed a clear reduction in biomass production under salinity stress. The average values of the absolute and relative growth rate intervals (AGR and RGR respectively) were significantly reduced under saline conditions in comparison with control conditions (Fig 2). For instance, the average RGR was reduced by 40% after 8–11 days under saline conditions compared to controls (Fig 2).

Similarly, the means of all agronomic traits measured in the field were significantly reduced under saline conditions compared with the controls (Table 1). For example, yield was reduced by 44.3%. We observed that the correlation between traits measured in the field varies, to some extent, according to treatment (S4 Fig). In our previous study, where a NAM population of barley was grown under same field conditions, we observed that a correlation between maturity time and ripening period became weaker under saline conditions [46]. Here, we found that maturity time and ripening period had a correlation of -0.48 under control conditions, but were not correlated under saline conditions. Similarly, harvest index and ripening period were correlated under control (0.36) but the correlation is not significant under saline

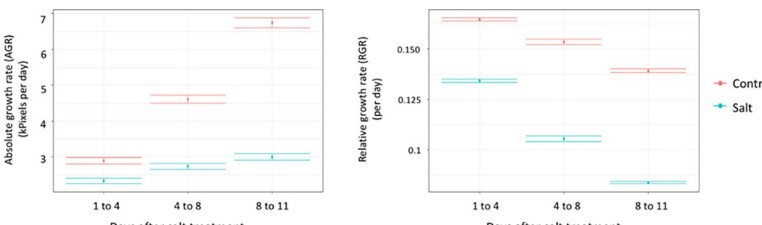

**Fig 2. The effect of salt treatment on the traits measured at The Plant Accelerator®.** Effect of salt on average absolute growth rate (AGR, left panel) and relative growth rate (RGR, right panel) for the three intervals 1–4, 4–8, and 8–11 days after salt treatment. Red and blue represent control and salt conditions, respectively. Error bars are ± standard error.

**Table 1. Effect of the salt treatment on agronomic traits measured in the field.**

| Trait [a] | Condition | Mean | SE [b] | CV (%) [c] |
|---|---|---|---|---|
| HEA (d) | Control | 85.1 | 0.267 | 8.61 |
| | Salt | 82 | 0.241 | 8.06 |
| MAT (d) | Control | 113 | 0.184 | 4.45 |
| | Salt | 108 | 0.209 | 5.31 |
| RIP (d) | Control | 28.3 | 0.124 | 12 |
| | Salt | 26.2 | 0.107 | 11.20 |
| HEI (cm) | Control | 67.3 | 0.383 | 15.6 |
| | Salt | 57.6 | 0.258 | 12.3 |
| EAR | Control | 4.34 | 0.0363 | 23.0 |
| | Salt | 2.68 | 0.0222 | 22.7 |
| GPE | Control | 14.6 | 0.102 | 19.1 |
| | Salt | 10.6 | 0.0802 | 20.7 |
| DRY_WT (g/m$^2$) | Control | 465 | 2.4 | 14.2 |
| | Salt | 375 | 2.36 | 17.3 |
| YLD (g/m$^2$) | Control | 155 | 1.4 | 24.7 |
| | Salt | 86.3 | 1.09 | 34.6 |
| HI | Control | 0.332 | 0.00215 | 17.8 |
| | Salt | 0.229 | 0.00233 | 28 |

Descriptive statistics (mean, standard error of the mean, and coefficient of variation) for the agronomic traits measured in the field under control and salt conditions.

a) Flowering time (HEA), maturity time (MAT), ripening period (RIP), plant height (HEI), ear number per plant (EAR), grain number per ear (GPE), dry mass per m$^2$ (DRY_WT), yield (YLD), and harvest index (HI)

b) SE refers to standard error of the mean

c) CV (%) refers to the percentage of the coefficient of variation (ratio of the standard deviation to the mean)

conditions. These results are similar to those reported in wheat grown in Indian fields, where variation was observed in the strength and significance of correlations among traits depending on the growing conditions (normal, saline, or sodic) [55].

## A locus is associated with a growth-related trait under controlled conditions independent of salt treatment

We used association mapping to identify genetic loci associated with the growth responses of barley to salinity stress during the vegetative stage. We applied the classical method using GAPIT [50] and the interaction method developed in Al-Tamimi *et al.* [24]. Using the classical model, we observed a significant association on chromosome 4H (112.65, 114.45 cM) for sodium and potassium contents (ionic phase) as measured by flame photometry in leaf samples collected from plants under saline conditions (S5 Table). The same locus was detected by Hazzouri *et al.* [12] and more recently by Houston *et al.* [13], the latter identifying *HKT1;5* as the gene underlying the association. This observation verifies that the genetic material and stress level applied at TPA were appropriate for the purpose of the experiment.

To identify loci related to stress tolerance and performance, the classical model was used on stress-tolerance indices derived for AGR and RGR from the three time-intervals (1–4, 4–8, and 8–11 days after salt treatment) in TPA. The classical model did not reveal any significant associations in this case and we could only find a weak, non-significant peak on chromosome 7H (44.35 cM) for RGR one to four days after the salt treatment (i.e., RGR1to4) using the stress tolerance index STI (as defined by Fernandez [56]) (S5 Fig). With the interaction model, we

detected the same locus on chromosome 7H (44.35 cM) associated with RGR1to4 independent of the treatment (S5 Fig). This locus was significantly associated with flowering time and plant height in a different barley population, grown under control conditions [57]. In this case, the interaction model detected QTL in the TPA better than the classical model, highlighting the importance of testing whether the markers underlying QTL associations significantly interact with treatment. Similarly, the interaction model detected QTL with much lower p-values than the classical method in a rice diversity panel tested for mild salinity stress [58]. Hence our results reinforce the need to employ the new models introduced in Al-Tamimi *et al*. [24] that include interactions in the association model rather than simply splitting the analysis between control and salt groups. No other significant peaks were identified, using either the classical model or the interaction model, for any of the other growth-related traits measured at TPA.

## Flowering loci and loci associated with yield components are responsive to salinity treatment under field conditions

In the association analyses of the field traits, loci with known underlying vernalization and flowering genes were significant under the marker and marker-by-treatment terms of the interaction model. A locus on chromosome 6H (51.93 cM) co-locating with *HvNAM-1* [59, 60] was significant under the marker-by-treatment term and associated with ripening period, plant height and ear number per plant (Fig 3). In addition, the 7H (28.46 cM) locus known to contain *Vrn-H3* (*HvFT1*) [61] was associated with flowering time, maturity time, and harvest index independent of the treatment. In contrast, the association of this locus with yield and ripening period was treatment-specific (Fig 3). In our diversity panel, the 5H (122.3 cM) locus containing *Vrn-H1* [62] affected ear number per plant depending upon whether the plant was grown under saline or control conditions.

The role of vernalization genes in barley response to salinity is debatable and is, we believe, either environmentally dependent or indirect. Zhou *et al*. [63] did not identify co-location among QTL underlying the vernalization genes (*Vrn-H1* and *Vrn-H3*) and those associated with salinity tolerance in a doubled-haploid barley population. However, a locus where *Vrn-H1* was associated with salinity tolerance under drained and waterlogged conditions was observed in another doubled haploid barley population [64]. In both studies [63, 64], salinity tolerance was assessed as a combination of leaf chlorosis and plant survival scores, and the plants were grown under glasshouse conditions. Results presented here suggest that the nature of the population and the method of stress evaluation might affect the apparent involvement of vernalization genes in salinity tolerance. The association mapping analyses presented here validate the results of our previous study that used a different barley population, and suggest that vernalization genes do play a role in the salinity stress response of barley under our field conditions [46].

We detected three other loci (2H, 141.87 cM; 3H, 139.22 cM; 7H, 128.35 cM) associated with yield components and response to salinity stress (Fig 3). The 2H (141.87 cM) locus was previously detected in a NAM wild barley population (2H, 140–145 cM) [46]. In the NAM population it was significant under saline conditions and was associated with yield [46]. *Alpha glucosidase*, *calreticulin 1* and *2*, and *choline-transporter-like* genes were suggested as potential candidates underlying this locus [46]. We detected the same locus (2H, 141.87 cM) but specifically associated with ear number per plant in response to salinity treatment. The identification of the same locus associated with yield and a yield-related trait in response to salt treatment in two distinct populations emphasizes its importance and justifies further investment in strategies to validate the causal gene.

We identified a QTL on chromosome 3H (139.22 cM) that was associated with ear number per plant in response to salinity treatment. To the best of our knowledge this QTL (139.22 cM)

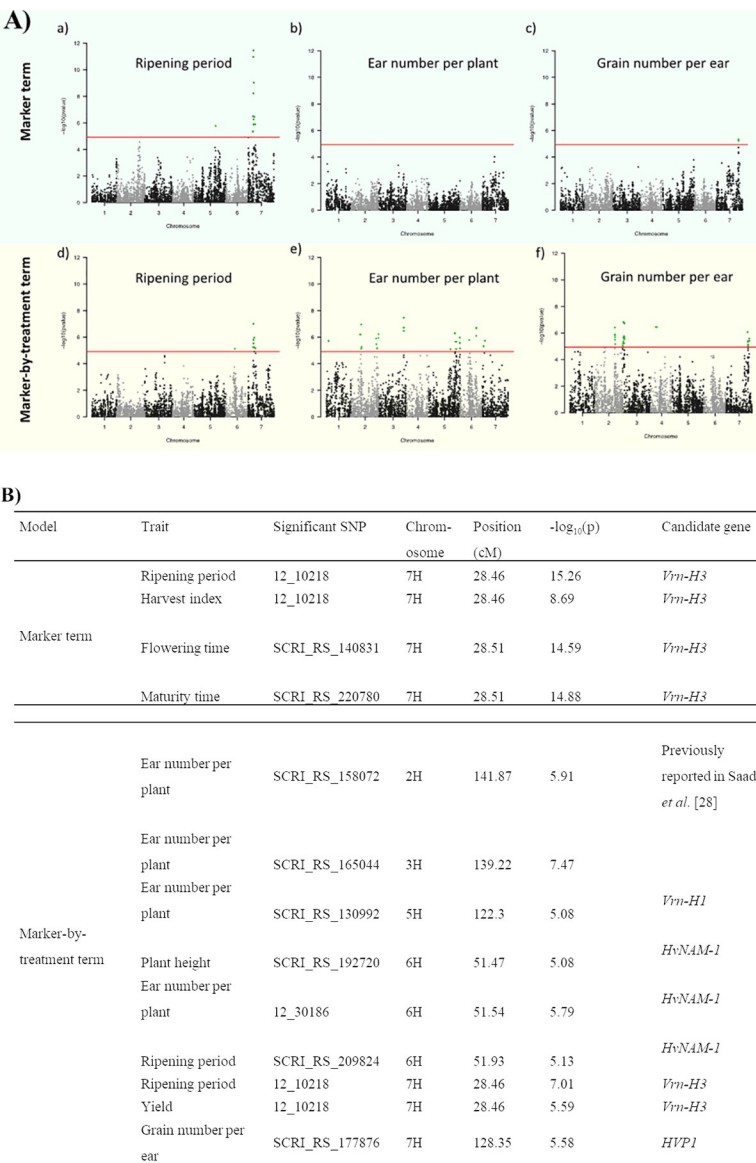

**Fig 3.** A) Manhattan plots of the interaction model for ripening period (a, d), ear number per plant (b, e), and grain number per ear (c, f). The upper and lower panels illustrate the marker and marker-by-treatment terms, respectively. Loci significant in the marker term contribute to the traits regardless of the treatment, whereas loci significant in the marker-by-treatment term are responsive to salinity treatment. The red line indicates the Bonferroni corrected p-value threshold above which SNPs are considered significant (indicated by green dots). B) List of loci discussed in the main text. These loci are significant under the interaction model for the marker or marker-by-treatment terms. The name, position, and–$\log_{10}$(p-value) of the most significant markers are mentioned and candidate genes are suggested.

has not previously been reported in either barley developmental or stress tolerance association studies. To explore the candidate genes underlying this QTL, we used the BARLEYMAP pipeline [65] with the updated Morex genome annotation [66], and searched for candidate genes within 2cM of the most significant marker (SCRI_RS_165044). We found that the candidate genes underlying the locus, a potassium transporter family protein (*HORVU3Hr1G104870*) and a glucan endo-1,3-beta-glucosidase (*HORVU3Hr1G105190*), were annotated as high-confidence genes within this genetic region. Both candidate genes were previously reported to be

involved in salinity responses. The ability to maintain a high cytosolic potassium-to-sodium ratio under salinity stress was found to be a key mechanism of salinity tolerance; hence the importance of potassium transporters [67]. In addition, glucan endo-1,3-beta-glucosidase, was shown to be responsive to salinity stress in different plant species. For example, the glucan endo-1,3-beta-glucosidase was downregulated by salt stress in the roots of tomato cultivars (*Solanum lycopersicum*) [68]. At the protein level, glucan endo-1,3-beta-glucosidase was induced by salt stress in the roots of upland cotton seedlings (*Gossypium hirsutum* L.) [69].

The third locus associated with a yield component (grain number per ear) under the marker-by-treatment term was located on chromosome 7H (128.35 cM). Interestingly, a pyro-phosphate-energized proton pump (*HORVU7Hr1G115540*, previously *AK360389*) was located at chromosome 7H position 128.68 cM. By blasting (http://webblast.ipk-gatersleben.de/barley_ibsc/) the sequence of *HVP1* mRNA for vacuolar H$^+$-translocating pyrophosphatase (GenBank: AB032839.1), we achieved the same hit (*HORVU7Hr1G115540*). Therefore, *HVP1* could be the candidate vacuolar H$^+$-translocating pyrophosphatase underlying the 7H locus. *HVP1* was previously mapped on chromosome 7H in a barley bi-parental population resulting from the Barque x CPI-71284 cross [70]; however, the low mapping resolution resulted in a large QTL with an imprecise position. The role of the vacuolar pyrophosphatase in salinity tol-erance was demonstrated in transgenic barley expressing the *Arabidopsis* vacuolar pyropho-sphatase *AVP1*, which showed increased shoot biomass production and yield under saline field conditions compared with the controls [71]. Therefore, *HVP1* merits further investigation as the causal gene underlying the 7H locus.

The means for ear number per plant and grain number per ear are presented in Table 2 for each condition (control or saline) and genotype (homozygous allele A or homozygous allele B at the most significant marker) at these three loci (2H, 141.87 cM; 3H, 139.22 cM; and 7H 128.35 cM). The condition, the genotype, and the condition-by-genotype interaction terms were all significant (two-way analysis of variance, p-value < 0.05), supporting these three loci as true positives. All three are therefore plausible candidates for new components of salinity tolerance in barley.

**Table 2. Comparison of trait (ear number per plant and grain number per ear) means between accessions by condition (control or saline) and genotype (at peak markers).**

| Trait | Peak marker | Chromosome | Position (in cM)[a] | Condition | Genotype[b] | Mean | Standard error |
|---|---|---|---|---|---|---|---|
| Ear number per plant | SCRI_RS_158072 | 2H | 141.87 | Control | AA | 4.39 | 0.04 |
| | | | | | GG | 3.83 | 0.14 |
| | | | | Saline | AA | 2.70 | 0.02 |
| | | | | | GG | 2.50 | 0.07 |
| Ear number per plant | SCRI_RS_165044 | 3H | 139.22 | Control | AA | 4.31 | 0.04 |
| | | | | | CC | 4.92 | 0.17 |
| | | | | Saline | AA | 2.68 | 0.02 |
| | | | | | CC | 2.80 | 0.10 |
| Grain number per ear | SCRI_RS_177876 | 7H | 128.35 | Control | AA | 14.39 | 0.12 |
| | | | | | CC | 15.35 | 0.19 |
| | | | | Saline | AA | 10.48 | 0.09 |
| | | | | | CC | 11.13 | 0.15 |

a) Position of peak marker in the QTL
b) Genotype at the peak marker

## Salinity tolerance in barley at the vegetative and reproductive stages seem to be controlled by distinct mechanisms

In this work, we sought to discover correlations between traits during the vegetative and reproductive stages of barley development. While we could not find a strong correlation between the traits measured in TPA and in the field, some traits were significantly correlated. For instance, we observed a correlation (r = 0.3–0.4) between dry mass per m$^2$ under control or saline conditions in the field, and different AGR intervals (Fig 4). We observed a similar correlation (r = 0.2–0.3) between the grain mass per m$^2$, i.e. yield, and different AGR intervals (Fig 4). This is consistent with observations that accessions that grew rapidly in TPA tended to have a large biomass and a higher yield in the field. Previous studies comparing data from controlled environment versus field studies have indicated that while certain correlations can be observed between the two experimental setups, they also report many inconsistencies [71, 72]. Our results, along with these previous observations highlight the need to exercise caution when attempting to extrapolate results collected in one environment, to another.

We could not identify common QTL between the TPA and field data. There are several possible reasons for this. Data from the two experimental setups showed only a modest correlation, potentially explaining the absence of common QTL. The barley plants studied in TPA

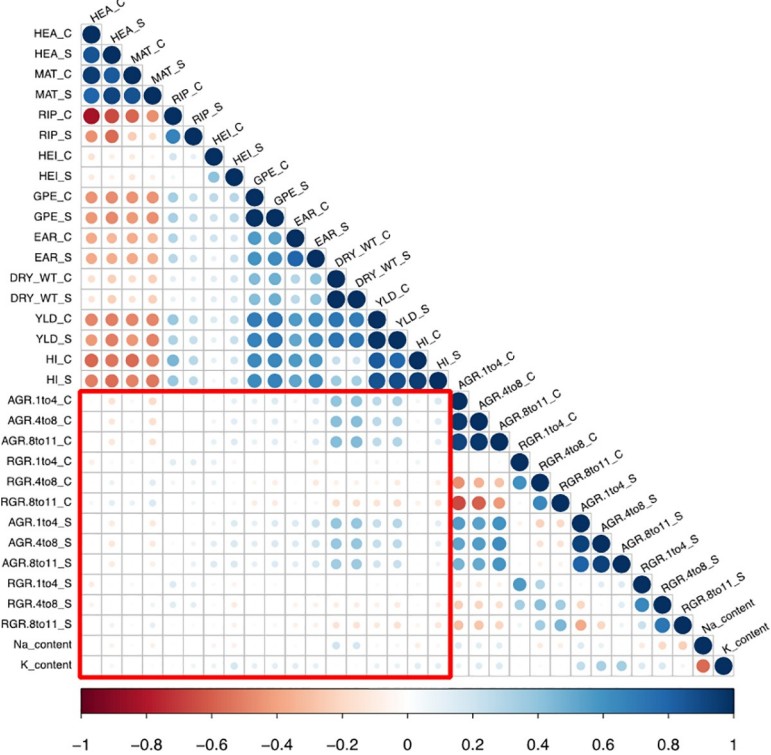

**Fig 4. Correlations among all traits measured under control and saline conditions in this study.** The red square encompasses correlations between the traits measured in the field at ICBA and under controlled conditions at TPA. The size and the color of the circles refer to the strength and the significance of the correlations, respectively. Nonsignificant correlations are indicated by blank cells. Flowering time (HEA), maturity time (MAT), ripening period (RIP), plant height (HEI), ear number per plant (EAR), grain number per ear (GPE), dry mass per m$^2$ (DRY_WT), yield (YLD), and harvest index (HI) are measured in the field under both control (_C) and saline (_S) conditions at ICBA. Absolute growth rate (AGR) and relative growth rates (RGR) are measured for the time-intervals 1–4, 4–8, and 8–11 days after salt treatment under both control (_C) and saline (_S) conditions at TPA. Sodium (Na) and potassium (K) contents were measured by flame photometer from leaves collected from plants grown at TPA under saline conditions.

were based on three-week-old plants, while those measured in the field were from plants in later developmental stages (mainly reproductive). Mano *et al.* [73] were unable to find overlapping QTL when studying barley during the germination and seedling stages, arguing that different tolerance mechanisms could be involved in the different stages. QTL that contribute to a certain trait can have a transient nature and dynamically change with time and developmental stages [54, 74, 75].

We detected only one significant QTL in TPA experimental setting (for RGR1to4). Two previous studies evaluated barley for water-deficit tolerance using high-throughput imaging and were able to detect QTL associated with growth rate [40, 76]. Both of these studies calculated growth differently from our study, with one fitting the logistic curve to describe plant growth [40, 76]. Al-Tamimi *et al.* [24] derived growth-related traits similarly to our study and identified QTL associated with RGR and AGR; however, the rice plants used were bigger than our plants (based on image comparison) and the authors used a 700k SNP chip for their analysis. The fact that many loci contributed to growth under saline conditions [24] was attributed to the complexity of the trait and the involvement of several loci with small effects. In the end, the paper was focused on QTL associated with transpiration efficiency [24]. Similarly, Peirone *et al.* [77] found that transpiration efficiency measured in greenhouse conditions at the early stages of soybean growth was the most efficient trait in predicting drought tolerance in the field and Knoch *et al.* [75] showed that early plant growth of canola was a complex trait that involves several temporally dynamic loci with medium and small effects. Furthermore, the heritabilities of TPA-measured traits were relatively low in our experiments, ranging between 0.5 and 0.6 (S6 Table), and these values were based on spatially adjusted means. Since previous high-throughput phenotyping experiments showed that the heritability values of growth-related traits become higher towards maturity [72], we suggest evaluating barley plants in TPA during a later developmental stage to better identify the growth-related loci that interact with salinity stress. Moreover, the advent of the barley 50k iSelect SNP chip [78] might improve the power of QTL detection. Finally, a model that runs the association analysis while including the spatial design terms in one step, rather than using the estimate of the means, might improve the association and enable the identification of significant QTL for salinity tolerance at TPA.

In this study, we identified new components of salinity tolerance—maintenance of grain number per ear and ear number per plant—to be added to the three components of salinity tolerance in the conventional paradigm of Munns *et al.* [3]. Furthermore, we provided some suggestions for improvement of the experiments performed at TPA in order to detect growth-related QTL associated with salinity tolerance in barley. Finally, we underscored the importance of validating the 2H (141.87 cM) locus and confirming *HVP1* as the candidate gene underlying the 7H (128.35 cM) locus.

## Supporting information

**S1 Fig. Results of STRUCTURE software ran on the 377 barley accessions used in our study.** a) We ran STRUCTURE for K from 1 to 6. Each vertical bar represents an accession. The labels from closest to furthest from the plot are year of release, country of origin, and name of the 377 accessions. b) *K* = 2 was chosen as the best *K* based on the *ΔK* method [33] using HARVESTER STRUCTURE software. c) PC1 and PC2 generated by GAPIT are plotted and colored by the year of release of the accession. The year 1991 was used as the cutoff to separate "old" from "new" accessions in this graph. At the bottom, the result from the STRUCTURE software for *K* = 2 is shown, ordered by year of the accession's release.
(TIF)

**S2 Fig. Experimental design at The Plant Accelerator®.** In top panels, allocation of replicated (blue), unreplicated (grey) accessions. In bottom panels, allocation of control (blue, 1) and salt (yellow, 2) conditions. The design is a split-plot design where each pair of carts corresponds to a main plot to which an accession is allocated. The left and right panels correspond to the design in the Northwest (NW) and Northeast (NE) smarthouses, respectively. Each position in a smarthouse corresponds to an individual plant.
(TIF)

**S3 Fig. Manhattan plots for the results of the interaction association model.** Results for flowering time (a, d), maturity time (b, e), plant height (c, f), dry mass per $m^2$ (g, j), yield per $m^2$ (h, k), and harvest index (i, l). Green- and yellow-shaded panels illustrate the marker and marker-by-treatment models, respectively. Loci significant in the marker term contribute to the traits regardless of the treatment, whereas loci significant in the marker-by-treatment term are responsive to salinity treatment. The red line indicates the Bonferroni corrected p-value threshold above which SNPs are significant (indicated by green dots).
(TIF)

**S4 Fig. Correlations between traits in the field under control (left) and salt (right) conditions.** The size and the color of the circles refer to the strength and the significance of the correlation, respectively. Non-significant correlations are indicated by blank cells.
(TIF)

**S5 Fig. Comparing Manhattan plots of Relative Growth Rate (RGR) for the interval 1–4 days (RGR1to4) after salt treatment measured at The Plant Accelerator® using different association models.** Manhattan plots of RGR 1to4 for a) control using GAPIT b) salt using GAPIT c) stress tolerance index STI (as defined in Fernandez [56]) using GAPIT d) marker term of interaction model e) marker-by-treatment term of interaction model. Loci significant in the marker term contribute to the traits regardless of the treatment, whereas loci significant in the marker-by-treatment term are responsive to salinity treatment.
(TIF)

**S1 Table. List of the 377 accessions used in this study, their country of origin and their year of release.**
(XLSX)

**S2 Table. Spatially corrected phenotypic data and the three principal components used for association analysis.**
(XLSX)

**S3 Table.** Results of the association analyses using the interaction model of the traits measured at a) The Plant Accelerator® and b) in the field.
(XLSX)

**S4 Table. LD (r2) between each pair of markers calculated using PLINK and split per chromosome.**
(XLSX)

**S5 Table. GAPIT results for sodium and potassium content per dry mass of leaf from plants grown at The Plant Accelerator® under salt stress.**
(XLSX)

**S6 Table. Heritability of the traits measured at The Plant Accelerator®.**
(XLSX)

## Acknowledgments

The research reported in this publication was supported by funding from King Abdullah University of Science and Technology (KAUST) to the baseline of MT. We thank Dr. Ibrahim S. Elbasyoni for helping design the field experiments. We thank the International Center for Biosaline Agriculture for their work in performing the field experiments. We acknowledge the Association Genetics of UK Elite Barley (AGOUEB) consortium for putting together the association mapping population. We would also like to extend our gratitude to the team members at The Plant Accelerator®, who provided assistance with the phenotypic data collection. The Plant Accelerator®, Australian Plant Phenomics Facility, is supported under the National Collaborative Research Infrastructure Strategy (NCRIS) of the Australian Government.

## Author Contributions

**Conceptualization:** Sónia Negrão, Mark Tester.

**Data curation:** Stephanie Saade, Chris Brien.

**Formal analysis:** Stephanie Saade, Chris Brien, Yveline Pailles.

**Funding acquisition:** Mark Tester.

**Investigation:** Stephanie Saade.

**Methodology:** Stephanie Saade, Chris Brien, Yveline Pailles, Bettina Berger, Mohammad Shahid.

**Project administration:** Mark Tester.

**Resources:** Joanne Russell, Robbie Waugh, Mark Tester.

**Supervision:** Sónia Negrão, Mark Tester.

**Validation:** Stephanie Saade.

**Writing – original draft:** Stephanie Saade.

**Writing – review & editing:** Stephanie Saade, Chris Brien, Bettina Berger, Joanne Russell, Robbie Waugh, Sónia Negrão, Mark Tester.

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
