## [Decision Letter · Decision Letter 0]

7 May 2020

PONE-D-20-04581

Dissecting new genetic components of salinity tolerance in two-row spring barley at the vegetative and reproductive stages

PLOS ONE

Dear Dr Saade,

Thank you for submitting your manuscript to PLOS ONE. After careful consideration, we feel that it has merit but does not fully meet PLOS ONE’s publication criteria as it currently stands. Therefore, we invite you to submit a revised version of the manuscript that addresses the points raised during the review process.

The paper is not written in the standard format for PLOS ONE. Please make changes in your revision. Please read both reviewers' comments and address the points raised. 

We would appreciate receiving your revised manuscript by 8 June 2020. To enhance the reproducibility of your results, we recommend that if applicable you deposit your laboratory protocols in protocols.io, where a protocol can be assigned its own identifier (DOI) such that it can be cited independently in the future. For instructions see: http://journals.plos.org/plosone/s/submission-guidelines#loc-laboratory-protocols

We look forward to receiving your revised manuscript.

Kind regards,

Chengdao Li, PhD

Academic Editor

PLOS ONE

Reviewers' comments:

Reviewer's Responses to Questions

**Comments to the Author**

1. Is the manuscript technically sound, and do the data support the conclusions?

Reviewer #1: Yes

Reviewer #2: Yes

2. Has the statistical analysis been performed appropriately and rigorously? 

Reviewer #1: No

Reviewer #2: Yes

3. Have the authors made all data underlying the findings in their manuscript fully available?

Reviewer #1: No

Reviewer #2: Yes

4. Is the manuscript presented in an intelligible fashion and written in standard English?

Reviewer #1: Yes

Reviewer #2: Yes

5. Review Comments to the Author

Reviewer #1: In the manuscript entitled “Dissecting new genetic components of salinity tolerance in two-row spring barley at the vegetative and reproductive stages”, the authors performed GWAS for salt-tolerance related traits using 377 spring barley accessions, attempting to identify genetic components of barley salinity tolerance at two stages. Definitely, the study provides useful information of major QTLs related to salt tolerance in barley.

However, in the MS, there are many issues should be definite and addressed.

In my opinion, in the MS, the phenotypic data of traits at the reproductive stage was not well organized. For example, how about the correlation of the yield traits under salinity field condition between the seasons 2013/2014 and 2014/2015? If the correlation of the trait is not good between the two years, it will influence the results of the GWAS based on average value of these traits.

For the experiment design, the time interval (within 11 days after 200 mM salt treatment in the soil pot) may be not enough to show the difference among accessions. Is it a reason to explain why no common QTLs is detected between the two stages ? More discussion would be addressed in the part of discussion.

The figure 1 is not easy to understand. How to reflect the genotypic difference of the effect of salt treatment on the traits in this figure?

In the legend of figure 2, the authors did not explain marker term and marker-by- treatment term.

In figure 3, there was no significant correlation between ion content and other traits. In the experiment, the fourth complete leaf was used to analyze Na and K content at the vegetative stage, why choose the fourth complete leaf for analysis?

Minor comments: the notes of x-axis in the figure S2 is missing. Table S5 had the data of TGW (‘Thousand grain weight’ in short?), but this trait was not mentioned in the MS.

Reviewer #2: The authors presented an interesting paper with the aims to find new genetic components of salinity tolerance in barley at the vegetative and reproductive stages. They sreened a large population of barley cultivars using the high-throughput phenotyping platform, which is novel and very important for improving the efficiency and accuracy of the labour-intensive phenotyping process. I only have two minor suggestions.

1.) The paper contains lots of data, but these were mostly presented in the Supplementary Files. One would suggest that a couple of these figures/tables should be presented in the paper. This will be useful for the readers to understand the high-tech and highly efficient phenotyping platform for screening large number of lines of crops such as barley and wheat.

2.) The paper combines the Results with Discussion, which is no an issue for many journals. It does not seem to be a standard format for PLoS One. I leave this for the Editor to sort out.

6. PLOS authors have the option to publish the peer review history of their article (what does this mean?). If published, this will include your full peer review and any attached files.

Reviewer #1: No

Reviewer #2: No

---

## [Author Response · Author response to Decision Letter 0]

5 Jun 2020

We thank the reviewers for their comments, which we have replied to below.

Reviewer #1: In the manuscript entitled “Dissecting new genetic components of salinity tolerance in two-row spring barley at the vegetative and reproductive stages”, the authors performed GWAS for salt-tolerance related traits using 377 spring barley accessions, attempting to identify genetic components of barley salinity tolerance at two stages. Definitely, the study provides useful information of major QTLs related to salt tolerance in barley.

However, in the MS, there are many issues should be definite and addressed.

- In my opinion, in the MS, the phenotypic data of traits at the reproductive stage was not well organized. For example, how about the correlation of the yield traits under salinity field condition between the seasons 2013/2014 and 2014/2015? If the correlation of the trait is not good between the two years, it will influence the results of the GWAS based on average value of these traits.

Response: 

The data used to run the GWAS model is the spatial adjusted data for site (assuming 4 sites: 2013_control, 2013_salt, 2014_control, 2014_salt) as described in the Materials and methods section Lines 291-293 and 295-297.

The correlations between 2013_control - 2014_control and 2013_salt - 2014_salt are high (a bit too high as they are inflated due to the spatial adjustment per site).

 Pearson correlation coefficient between years

Trait Control Salt Both treatments

HEA 0.8407097 0.932204 0.87892784

MAT 0.8662038 0.904021 0.89078656

RIP 0.6053287 0.704846 0.67635931

HEI 0.9999970 0.999999 0.99224917

EAR 0.9684144 0.797763 0.9669035

GPE 0.9498838 0.951435 0.92541853

DRY_WT 0.8855445 0.937100 0.90577249

YLD 0.9938078 0.995805 0.95926533

HI 0.8951714 0.967731 0.84466089

We added a sentence “, using predicted means adjusted for year and condition.” on line 304 to re-emphasize that.

- For the experiment design, the time interval (within 11 days after 200 mM salt treatment in the soil pot) may be not enough to show the difference among accessions. Is it a reason to explain why no common QTLs is detected between the two stages? More discussion would be addressed in the part of discussion.

Response: The ionic phase of salt response is due to the accumulation of the sodium ions in the shoot. The shoot ion-independent phase (also known as osmotic phase) happens within few hours to few days of the salt treatment [1]. The ionic phase is more extensively studied compared with the shoot ion-independent phase, where phenotyping is the bottleneck. Our aim was to use high-throughput phenotyping to study the shoot ion-independent phase (Lines 130-132). Therefore, 11 days is actually more than enough. However, we hypothesized in the manuscript (Lines 550-551, 563-564) that the plants were too young when the salt was imposed possibly making it hard to see differences among accessions. 

To address the reviewer’s comment and clarify that the shoot ion-independent phase of the plant response to salt stress is rapid and is manifested by growth reduction, we added the sentence “shoot ion-independent tolerance takes place in the early stage of the salt stress before sodium ions had the time to accumulate. During this rapid response of the plant to stress, growth is reduced” (Lines 96-98).

- The figure 1 is not easy to understand. How to reflect the genotypic difference of the effect of salt treatment on the traits in this figure?

Response: The aim of Fig 1 is to show that the salt treatment (200mM) 

is considered appropriate for our studies because it significantly affected all studied traits without causing major premature plant senescence. As shown in Fig 1, there is significant difference between average of trait under control compared with average of trait under salt treatment. 

The sentence “and natural genetic variation across the population in the magnitude of this response.” at Line 342 was not referring to Fig 1. We deleted it to avoid confusion.

- In the legend of figure 2, the authors did not explain marker term and marker-by- treatment term.

Response: we addressed this by explaining the terms in the legend (this is Fig 3 now). We also added the explanation of those terms to the legends of S3 and S5 Figs.

- In figure 3, there was no significant correlation between ion content and other traits. In the experiment, the fourth complete leaf was used to analyze Na and K content at the vegetative stage, why choose the fourth complete leaf for analysis?

Response: (This is Fig 4 now) Choosing the first or second fully expanded leaf is a standard practice in salinity stress evaluation. This ensures that the leaves collected from varying genotypes have been exposed to the same period of treatment. 

Traits measured at The Plant Accelerator® are growth related (absolute growth rate and relative growth rate) and leaf Na and K contents. The former correlates with the shoot ion-independent phase of the response to salt treatment while the latter correlates with the ionic phase. This explains why Na and K content did not significantly correlate with the other traits at The Plant Accelerator®. Na content usually correlates with senescence as the plant response to salt stress. As mentioned in Line 383-385, detecting the known locus on chromosome 4H for sodium and potassium contents proves that the genetic material is appropriate for this study.

To address the reviewer’s comment, we added “This ensured that the collected leaf had been exposed to the same period of stress despite variations in plant growth across accessions” (Lines 199-201) to the Materials and methods section to explain that choosing the first or second fully expanded leaf is a standard practice in salinity stress evaluation.

We also re-emphasized in the Discussion section that Na and K content are related to the ionic phase (Line 380). 

- Minor comments: the notes of x-axis in the figure S2 is missing. 

Response: We fixed this by adding the “chromosome” label, which was missing from the x-axis of the lower panel. 

- Table S5 had the data of TGW (‘Thousand grain weight’ in short?), but this trait was not mentioned in the MS.

Response: Data on TGW trait was initially collected. However, the trait had very low heritability and was not reliable so we decided not to include it in the manuscript. We kept the phenotypic data (Table S2) and GWAS results (Table S3 b, field) of the TGW (an important yield component) for transparency and future reference for readers.

Reviewer #2: The authors presented an interesting paper with the aims to find new genetic components of salinity tolerance in barley at the vegetative and reproductive stages. They sreened a large population of barley cultivars using the high-throughput phenotyping platform, which is novel and very important for improving the efficiency and accuracy of the labour-intensive phenotyping process. I only have two minor suggestions.

1.) The paper contains lots of data, but these were mostly presented in the Supplementary Files. One would suggest that a couple of these figures/tables should be presented in the paper. This will be useful for the readers to understand the high-tech and highly efficient phenotyping platform for screening large number of lines of crops such as barley and wheat.

Response: 

We addressed this by:

- Moving the table that compares trait (ear number per plant and grain number per ear) means between accessions by condition (control or saline) and genotype (at peak markers) from supplementary information to the main text as Table 2. 

- Moving the plots of smoothed absolute growth rate and relative growth rates at The Plant Accelerator® from supplementary information to the main text as Fig 1.

2.) The paper combines the Results with Discussion, which is no an issue for many journals. It does not seem to be a standard format for PLoS One. I leave this for the Editor to sort out.

Response: We understand that combining the Results with Discussion is not a standard format. However, it is not uncommon. We think that combining the results and discussion makes this paper concise, especially that results from both the field and The Plant Accelerator® need to be presented and discussed. In addition, the guidelines (https://journals.plos.org/plosone/s/submission-guidelines#loc-results-discussion-conclusions) state “Results, Discussion, Conclusions. These sections may all be separate, or may be combined to create a mixed Results/Discussion section (commonly labeled “Results and Discussion”) or a mixed Discussion/Conclusions section (commonly labeled “Discussion”).” Similarly, https://journals.plos.org/plosone/s/file?id=4497/Main%20Body%20-%20ONE%20Formatting.pdf show the results and discussion section together. Therefore, we prefer to keep the Results and discussion combined.

1. Munns R, Tester M. Mechanisms of salinity tolerance. Annual Review of Plant Biology. 2008;59:651-81. doi: 10.1146/annurev.arplant.59.032607.092911. PubMed PMID: WOS:000256593200026.

---

## [Editor Report · Decision Letter 1]

29 Jun 2020

Dissecting new genetic components of salinity tolerance in two-row spring barley at the vegetative and reproductive stages

PONE-D-20-04581R1

Dear Dr. Saade,

We’re pleased to inform you that your manuscript has been judged scientifically suitable for publication and will be formally accepted for publication once it meets all outstanding technical requirements.

Kind regards,

Chengdao Li, PhD

Academic Editor

PLOS ONE
---

## [Editor Report · Acceptance letter]

6 Jul 2020

PONE-D-20-04581R1 

Dissecting new genetic components of salinity tolerance in two-row spring barley at the vegetative and reproductive stages 

Dear Dr. Saade:

I'm pleased to inform you that your manuscript has been deemed suitable for publication in PLOS ONE. Congratulations! Your manuscript is now with our production department. 

Kind regards, 

on behalf of

Professor Chengdao Li 

Academic Editor

PLOS ONE